# Mechanistic Insights from Transcriptomics: How the Glucose Transporter *gltp1* Gene Knockout Enhances *Monascus* Pigment Biosynthesis in *M. ruber* CICC41233

**DOI:** 10.3390/jof11120867

**Published:** 2025-12-07

**Authors:** Chuannan Long, Qinqin Tao, Xinyi Liu, Jingjing Cui

**Affiliations:** 1School of Life Science, Jiangxi Science & Technology Normal University, Nanchang 330013, China; t1321136641@163.com (Q.T.); lxy200368@163.com (X.L.); 2Analysis and Testing Center, Jiangxi Science & Technology Normal University, Nanchang 330013, China; 3Key Laboratory of Natural Microbial Medicine Research of Jiangxi Province, Jiangxi Science & Technology Normal University, Nanchang 330013, China; 4Jiangxi Provincal Key Laboratory of Organic Functional Molecules, Institute of Organic Chemistry, Jiangxi Science & Technology Normal University, Nanchang 330013, China

**Keywords:** *Monascus ruber*, glucose transporter GLTP1, *Monascus* pigment, gene knockout, transcriptome

## Abstract

This study’s objective was to evaluate the effect of the glucose transporter GLTP1 in *Monascus ruber* CICC41233 on *Monascus* pigment biosynthesis. The *gltp1* gene in *M. ruber* CICC41233 was cloned to construct the overexpression vector pNeo0380-gltp1, resulting in complementation and overexpression strains, and its upstream and downstream homologous arms were used to construct the gene knockout plasmid pHph0380G/Gltp1::hph, resulting in a mutant strain. The results showed that the *gltp1* gene knockout strain *M. ruber* GLTP24 exhibited dramatically accelerated starch degradation and a significant increase (74.1% higher) in the yield of alcohol-soluble pigments compared to the wild-type. Reverse genetic experiments confirmed this phenotype: complementation strains restored wild-type pigment production levels, while overexpression strains showed reduced pigment synthesis. Integrated transcriptomic analyses revealed that *gltp1* deletion triggered extensive metabolic reprogramming. This included the downregulation of key components in the carbon-sensing GprD-cAMP/PKA signaling pathway and the concerted upregulation of multiple amino acid metabolic pathways, which supply essential precursors and amino groups for *Monascus* pigment synthesis. This study provides novel insights into the molecular link between carbon transport, signaling, and *Monascus* pigments in *Monascus ruber*.

## 1. Introduction

Red mold rice (RMR), which is fermented from rice using *Monascus* spp., is a traditional food product with a history of over a thousand years in China [1,2]. It is widely used as a natural food colorant, preservative, and ingredient in the production of rice wine, vinegar, and meat products. Beyond its culinary applications, RMR is also recognized for its functional properties, including antihypertensive, lipid-lowering, antidiabetic, and anti-inflammatory effects, leading to its classification as functional RMR [2,3,4,5,6]. With an annual production of approximately 20,000 tons in China and over 100 million people consuming *Monascus*-derived products daily [6], research into the high-efficiency synthesis mechanisms of *Monascus* pigments (MPs) has gained significant importance.

*Monascus* pigments are a mixture of compounds and are typically classified into the following three groups based on their absorption spectra: yellow (330–450 nm), orange (460–480 nm), and red (490–530 nm) pigments [6,7]. According to solubility, they can be divided into water-soluble and alcohol-soluble forms, with the latter being the predominant intracellular form [6]. They are secondary metabolites synthesized via the polyketide pathway, using acetyl-CoA as a precursor [8]. The biosynthesis pathway involves key enzymes such as polyketide synthase (PKS) and fatty acid synthase (FAS) [8]. Recent genomic studies have identified a gene cluster spanning approximately 53 kb responsible for MP biosynthesis, which includes genes encoding PKS, FAS, regulatory proteins, and transporters [1].

Nitrogen sources, particularly amino acids, play a critical role in regulating MP biosynthesis and composition. It is proposed that the orange pigment is directly synthesized via the polyketide pathway, the yellow pigment is formed by the reduction in the orange pigment using reducing equivalents (e.g., NADPH), and the red pigment is generated through the amination of the orange pigment, utilizing amino groups derived from amino acid metabolism [5,6]. Specific amino acids have been shown to differentially influence the pigment profile. For instance, supplementation with phenylalanine, valine, leucine, and isoleucine increased the proportion of yellow and orange pigments in *Monascus* sp. KCCM 10093, while serine, histidine, glycine, and alanine favored the production of red pigments [9]. Similarly, histidine, glycine, arginine, tyrosine, and serine significantly promoted red pigment formation in *Monascus ruber* ATCC 96218 [10], with arginine suggested as a key amino group donor [6]. Beyond specific amino acids, inorganic nitrogen sources like ammonium sulfate can also alter pigment composition, increasing the proportion of intracellular red pigments while decreasing yellow pigments [11]. These findings underscore the intricate interplay between nitrogen metabolism and MP synthesis.

Numerous studies have explored strategies to enhance MP production by modulating central metabolic pathways. In *Monascus ruber* CICC41233, heterologous expression of the α-amylase gene AOamyA from *Aspergillus oryzae* resulted in the engineered strain *Monascus ruber* Amy9. When fermented with rice as the substrate, starch was completely degraded in the Amy9 strain after 2 days, whereas the wild-type strain still retained residual starch even after 6 days of fermentation. After 6 days of fermentation, the yield of *Monascus* pigments in the engineered strain *M. ruber* Amy9 increased by 132% compared to the wild-type strain [12]. Furthermore, overexpression of the gene encoding the endogenous α-amylase MrAMY1 in *M. ruber* CICC41233 (which shares up to 69% homology with AOamyA) led to complete starch degradation in the resulting engineered strain within 2 days. After 6 days of fermentation, the pigment color value increased by 71.69% compared to the wild-type strain [13]. These results demonstrate a positive correlation between the rapid degradation of starch and the production of *Monascus* pigments. Overexpression of ATP-citrate lyase (ACL) increases acetyl-CoA supply and boosts pigment yield [14]. Modulation of fatty acid metabolism, such as knocking out the ergosterol biosynthesis gene *ERG4* or overexpressing acyl-CoA binding proteins (ACBPs), redirects metabolic flux toward pigment synthesis [15,16,17]. Furthermore, key regulatory elements, including the global regulator *LaeA* [18], the carbon catabolite repressor *CreA* [5], and G-protein signaling components (e.g., *mga1*, *flbA*), have been identified as critical modulators of MP biosynthesis [19,20,21].

Our preliminary results indicated that *M. ruber* CICC41233 produced substantial amounts of *Monascus* pigments after 2 days of fermentation using rice as the substrate, whereas no pigment production was observed when glucose was used as the carbon source under the same conditions. Transcriptome sequencing revealed that among the major facilitator superfamily (MFS) monosaccharide transporters, only one glucose transporter (designated GLTP1), encoded by the gene *gltp1*, was significantly upregulated. The expression level (FPKM values) of *gltp1* was 4506.55 in rice-based fermentation and 1389.04 in glucose-based fermentation (unpublished data). In the *M. ruber* NRRL1597 genome database (https://mycocosm.jgi.doe.gov/Monru1/Monru1.home.html) (accessed on 20 September 2016), this transporter also exhibited 67% homology with glucose transporters from *Aspergillus* species, such as *Aspergillus fumigatus* Af293 (Afu2g11520) and *Aspergillus nidulans* FGSC A4 (AN5860). These proteins belong to the low-affinity glucose transporter family, which enables rapid sensing of high glucose concentrations in the environment.

Therefore, this study aims to elucidate the molecular mechanism by which GLTP1 influences *Monascus* pigment synthesis in *M. ruber*. Using genetic approaches including gene knockout, complementation, and overexpression, combined with transcriptomic analyses, we seek to uncover the regulatory network connecting glucose transport and pigment biosynthesis. This research will not only advance our understanding of the crosstalk between primary and secondary metabolism in *M. ruber*, but also provide a theoretical foundation for enhancing the industrial production of *Monascus* pigments.

## 2. Materials and Methods

### 2.1. Strains and Culture Conditions

The wild-type strain *M. ruber* CICC41233 was obtained from the China Center of Industrial Culture Collection. Cultivation of both wild-type and engineered strains was carried out on malt–peptone–starch (MPS) agar at 30 °C for 7 days [12,13,14,15]. For phenotypic characterization, a medium of MPS agar containing 0.2% acetate was utilized. Additionally, *Escherichia coli* DH5α and *Agrobacterium tumefaciens* EHA105 were used in accordance with previous methodologies [12,13,14,15].

### 2.2. Construction of gltp1 Gene Expression and Knockout Vectors and Transform

The *gltp1* gene (https://mycocosm.jgi.doe.gov/Monru1/Monru1.home.html, Protein ID 378211, gene_129444, accessed on 20 September 2016) fragment was amplified by PCR with the DNA of *M. ruber* CICC41233 as a template, and its sequence was analyzed in NCBI Nucleotide BLAST (https://blast.ncbi.nlm.nih.gov/Blast.cgi, accessed on 20 September 2016).

The *gltp1* gene fragment and the expression plasmid pNeo0380 [12,13,14,15] were digested with *Hin*dIII and *Sac*I, respectively. They were used to construct the expression vectors pNeo0380-gltp1.

The T-DNA binary vector pHph0380G/Gltp1::hph was constructed on the backbone of pHph0380G to knockout the *gltp1* gene. The first homologous arm sequence (fragment I = 1740 bp) of *gltp1* upstream was amplified by PCR with the primers G129444-QC-UF-*Pst*I (containing *Pst*I) and G129444-QC-UR-SacI (containing *Sac* I), and then it was inserted into the *Pst*I/*Sac* I sites of pHph0380G-GLTP. Then, the second homologous arm sequence (fragment II = 1764 bp) of *gltp1* downstream was amplified by PCR with the primers G129444-QC-DF-BglII (containing *Bgl*II) and G129444-QC-DR-SpeI (containing *Spe*I), and then it was also inserted into the *Bgl*II/*Spe*I sites of pHph0380G-GLTP. The primer sequences used in this study are listed in Appendix A and the schematic maps of the primer locus for the knockout of *gltp1* are shown in Figure 1A.

Subsequently, *A. tumefaciens* EHA105 harboring the vectors was used to transform *M. ruber* CICC41233. For overexpression of the *gltp1* gene, geneticin (80 μg/mL) was used as the selection agent, while hygromycin B (100 μg/mL) was used for the selection of *gltp1* knockout mutants [5]. Putative positive transformants in both cases were verified using previously established methods [12,13,14,15].

### 2.3. Analysis of MPs’ Production and Starch Content

MPs’ production was carried out in 250 mL Erlenmeyer flasks containing 50 mL of medium (composed of 9.0% rice powder, 0.2% NaNO_3_, 0.1% KH_2_PO_4_, 0.2% MgSO_4_·7H_2_O, and 0.2% acetate, pH 3.2). The flasks were inoculated with freshly harvested spores at a final density of 10^5^ conidia/mL and incubated at 30 °C with shaking at 180 rpm. Pigment yields were determined after 36, 48, and 144 h of fermentation. Following fermentation, extracellular and intracellular pigments, along with biomass, were analyzed according to established methods [12,13,14,15]. All experiments were independently performed in triplicate.

The absorbance spectrum scan of pigments was recorded by a UV/VIS spectrophotometer (Cary 60, AgilentTechnologies, Santa Clara, CA, USA) from 200 to 700 nm. The total MPs included extracellular and intracellular pigments.

Following fermentation, the starch content in the culture supernatant was determined using an iodine solution (2.6 g/L I_2_, 5.0 g/L KI).

### 2.4. High-Performance Liquid Chromatography (HPLC) Analysis

The glucose content in the culture supernatant was analyzed by high-performance liquid chromatography (HPLC) using a Waters e2695 (Waters Corporation, Milford, MA, USA) system equipped with an Aminex HPX-87H column (Bio-Rad, Hercules, CA, USA). The analysis was performed with a 5 mM H_2_SO_4_ mobile phase at a flow rate of 0.6 mL/min. A refractive index detector, maintained at 63 °C, was used for detection [14].

### 2.5. RNA Extraction and Transcriptome Sequencing

Following fermentation, 25 mL of the culture broth from both *M. ruber* CICC41233 and the Δ*gltp1* mutant was allocated for pigment analysis. The remainder was centrifuged (9000× *g*, 30 min, 4 °C), and the resulting cell pellet was immediately flash-frozen in liquid nitrogen, after which a portion of the samples was used for RNA extraction [12] and another for transcriptome sequencing. Specifically, samples from 36 h and 144 h cultures were sent on dry ice to Kidio Biotechnology Corporation (Guangzhou, China) for library preparation and sequencing. The transcriptome raw data were submitted to the NCBI database.

### 2.6. Quantitative Real-Time PCR

Gene expression analysis was carried out by reverse transcription followed by relative quantification, normalized to the *actin* gene as an endogenous reference, based on an established protocol [12]. All the *Gltp1* genes, *gene_449307* (Abbreviated as *G449307*), *gene_472784* (Abbreviated as *G472784*), *gene_440889* (Abbreviated as G*440889*), *gene_440059* (Abbreviated as *G440059*), *gene_469204* (Abbreviated as G*469204*), *gene_468731* (Abbreviated as *G468731*), and *gene_387285* (Abbreviated as *G387285*), were selected based on transcriptome sequencing, and the gene sequence information was obtained from the genome of *Monascus ruber* (https://mycocosm.jgi.doe.gov/Monru1/Monru1.home.html) (accessed on 25 October 2018) and analyzed. The primers actin-YGF1 and actin-YGR1, Gltp1-YGF and Gltp1-YGR, G449307-YGF and G449307-YGR, G472784-YGF and G472784-YGR, G440889-YGF and G440889-YGR, G440059-YGF and G440059-YGR, G469204-YGF and G469204-YGR, G468731-YGF and G468731-YGR, and G387285-YGF and G387285-YGR were designed by software Primer Premier 5 to use to analyze *actin*, *Gltp1*, *G449307*, *G472784*, *G440889*, *G440059*, *G469204*, *G468731*, and *G387285* gene expression levels, respectively. Appendix A lists all the primer sequences employed in this study.

### 2.7. Statistical Analysis

All experiments were performed with at least three independent replicates. Data are presented as mean ± standard deviation (SD). Statistical significance among different treatment groups was determined by one-way analysis of variance (ANOVA) followed by Tukey’s test, with a *p*-value of less than 0.05 considered statistically significant.

## 3. Results and Discussion

### 3.1. Obtain the Δgltp1 Mutant Strain

The *gltp1* gene was cloned from *M. ruber* CICC41233, and its sequence was subjected to NCBI Nucleotide BLAST analysis. It showed the highest homology (97%) with the hexose transporter hxt1 (TQB68571.1) of *Monascus purpureus*.

Then, the *A. tumefaciens* EHA105 mediated transformation of pHph0380G/Gltp1::hph into *M. ruber* CICC41233, with nine transformants initially selected and confirmed to exhibit mitotic stability. Subsequent PCR analysis of genomic DNA from these candidates, however, demonstrated that only one was positive. Firstly, the primers G129444-JYF-HindIII (F3) and G129444-JYR-SacI (R3) were used to detect the *gltp1* gene (948 bp), which was present in the parental strain but was absent in the Δ*gltp1* mutant (Figure 1A,B). At the same time, the primers Hph-1034F (F4) and Hph-1034R (R4) were used to detect *hph* gene (1034 bp), which was absent in the parental strain but was present in the Δ*gltp1* mutant (Figure 1A,B). Then, the primers HtrpC-YZF (F5) and G129444-YZR (R5) were used to verify that the hph cassettes (2644 bp) were successfully inserted into the *gltp1* locus in the Δ*gltp1* mutant (Figure 1A,B). Therefore, the mutant was named *M. ruber* GLTP24. The phenotype of the parental strain *M. ruber* CICC41233 and mutant *M. ruber* GLTP24 were grown on MPS for 7 d, 8 d, and 9 d (Figure 1C). The Δ*gltp1* mutant grew denser than the parental strain (Figure 1C).

**Figure 1 jof-11-00867-f001:**
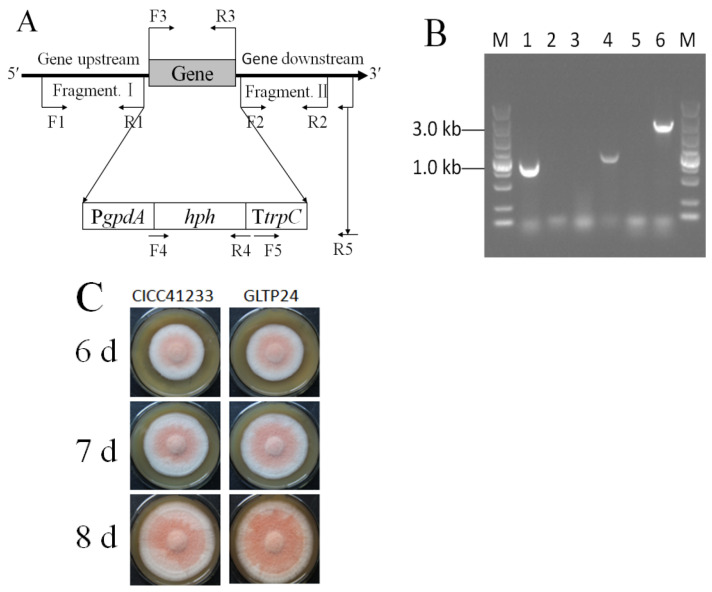
Schematic map of the primer locus for knocking out *gltp1* (**A**) and identification of mutant *M. ruber* GLTP24 by PCR (**B**) and phenotype of *M. ruber* (**C**). (**A**) F1, primer G129444-QC-UF-*Pst*I; R1, primer G129444-QC-UR-SacI. F2, primer G129444-QC-DF-BglII; R2, primer G129444-QC-DR-SpeI. F3, G129444-JYF-HindIII; R3, primer G129444-JYR-SacI. F4, primer Hph-1034F; R4, primer Hph-1034R. F5, primer HtrpC-YZF; R5, primer G129444-YZR. (**B**) The fragments of lanes 1 and 2 were products of the primers F3 and R3 test. The fragments of lanes 3 and 4 were products of the primers F4 and R4 test. The fragments of lanes 5 and 6 were products of the primers F5 and R5 test. Lanes 1, 3, and 5 represent the genomic DNA as the template from the parental strain *M. ruber* CICC41233; lanes 2, 4, and 6 represent the genomic DNA as the template from the mutaint strain *M. ruber* GLTP24. M: DL5000 bp DNA Ladder Marker. (**C**) Phenotype of *M. ruber* grown on MPS medium.

### 3.2. Construct Engineered Strains with the gltp1 Gene Complementation and Overexpression Strains

Then, the *gltp1* gene was homologously overexpressed in *M. ruber* CICC41233, and three transformants tested positive, named as *M. ruber* OE3, *M. ruber* OE4, and *M. ruber* OE7 (the molecular identification results were not presented). The *gltp1* gene was expressed in the mutant strain *M. ruber* GLTP24, and three complementary strains tested positive, named as *M. ruber* HU1, *M. ruber* HU 6, and *M. ruber* HU 17 (the molecular identification results are not presented).

### 3.3. Comparison of MPs’ Production in M. ruber CICC41233 and Mutant M. ruber GLTP24

The phenotypic appearance of *M. ruber* CICC41233 and *M. ruber* GLTP24 fermented using rice flour as the substrate for pigment production is shown in Figure 2A. At 36 h, the mutant strain yielded significantly higher amounts of visual *Monascus* pigments than the parental strain. The biomass production showed no significant difference between the strains.

More significantly, during fermentation with rice flour, the mutant strain displayed a dramatic acceleration in starch consumption. The residual starch content in the *M. ruber* CICC41233 fermentation samples at 36 h, 48 h, and 144 h was 44.91 mg/mL, 34.26 mg/mL, and 9.58 mg/mL, respectively. In contrast, *M. ruber* GLTP24 showed significantly lower levels, as follows: 0.56 mg/mL, 0.15 mg/mL, and 0 mg/mL, respectively (Figure 2B).

Concomitant with rapid starch degradation, the *M. ruber* GLTP24 mutant produced significantly higher yields of *Monascus* pigments (MPs). The pigment yield of *M. ruber* GLTP24 (49.46 U/mL) was significantly higher than that of *M. ruber* CICC41233 (28.42 U/mL) at 144 h. The results demonstrate that *M. ruber* GLTP24 can enhance the production of yellow and red pigments in the alcohol-soluble fraction of *Monascus* pigments (Figure 2C,E) [6]. This establishes a clear positive correlation between the rate of starch degradation and the yield of key MPs, a phenomenon supported by previous studies where accelerated starch hydrolysis led to increased pigment production [12,13]. However, the production of yellow and red pigments in the water-soluble fraction of *Monascus* pigments saw a slight decrease (Figure 2C,D).

As a low-affinity glucose transporter, GLTP1 can transport extracellular glucose. Consequently, knockout of the *gltp1* gene resulted in a significant difference in extracellular glucose concentration between *M. ruber* CICC41233 and *M. ruber* GLTP24 (Figure 2F). This indicates that the deletion of the *gltp1* glucose transporter paradoxically enhances the degradation of starch.

Our findings depict GLTP1 not merely as a glucose transporter, but as a critical metabolic regulator that couples carbon sensing with secondary metabolism. The observation that its deletion enhances starch degradation suggests that GLTP1 may be involved in a carbon catabolite repression (CCR) mechanism. In many fungi, efficient glucose uptake through specific transporters sustains CCR, repressing the expression of genes for utilizing alternative carbon sources (like starch) and the synthesis of secondary metabolites [5,22].

We propose a model wherein the knockout of *gltp1* disrupts this repression signal. The inability to efficiently transport glucose, despite its availability from starch hydrolysis, may be perceived by the cell as a carbon-limited state. This is consistent with the observed downregulation of carbon-sensing components like the G-protein-coupled receptor GprD and the protein kinase Pka-C3 in the mutant, which are part of a signaling cascade known to influence fungal development and secondary metabolism [21,22,23].

In conclusion, GLTP1 serves as a key metabolic gatekeeper. Its deletion deregulates carbon catabolite repression, leading to accelerated starch assimilation and a rechanneling of carbon flux towards the enhanced biosynthesis of *Monascus* pigments.

### 3.4. Comparison of MPs’ Production in M. ruber CICC41233 and Complementation and Overexpression Strains

To further verify the impact of the *gltp1* gene on *Monascus* pigment production in *M. ruber*, the functional role of *gltp1* was unequivocally confirmed by reverse genetics. The complementary strains (*M. ruber* HU1, *M. ruber* HU6, and *M. ruber* HU17), in which the *gltp1* gene was restored, showed recovery of the wild-type phenotype, with no significant difference in MPs’ production compared to *M. ruber* CICC41233 (Figure 3A). This demonstrates that the hyper-producing phenotype of the *M. ruber* GLTP24 mutant is directly attributable to the loss of *gltp1*.

Conversely, overexpressing *gltp1* led to a contrasting effect. The overexpression strains (*M. ruber* OE3, *M. ruber* OE4, and *M. ruber* OE7) consistently produced lower levels of alcohol-soluble and total pigments compared to the wild-type (Figure 3B). The expression levels of the *gltp1* gene are shown in Figure 4A. In comparison with *M. ruber* CICC41233, the expression fold of *gltp1* increased by 3.38-, 4.50-, and 2.84-fold in *M. ruber* OE3, *M. ruber* OE4, and *M. ruber* OE7 after 6 days, respectively. This inverse relationship between *gltp1* expression levels and pigment yield provides compelling evidence that GLTP1 acts as a negative regulator of MPs’ biosynthesis.

### 3.5. Global Transcriptional Reprogramming Induced by *gltp1* Deletion

The raw transcriptome data are available in the NCBI database under the accession number PRJNA1370451. The deletion of the *gltp1* gene in *M. ruber* triggered extensive transcriptional changes, as evidenced by the high number of differentially expressed genes (DEGs) when comparing the mutant (*M. ruber* GLTP24) to the wild-type (*M. ruber CICC41233*) at both 36 h (S01 vs. S02, 1249 DEGs) and 144 h (S03 vs. S04, 1041 DEGs) of fermentation (Table 1). qRT-PCR was performed to corroborate the transcriptomic data obtained from RNA-seq (Figure 4B). In both comparisons, the number of downregulated genes surpassed that of upregulated genes, indicating that the loss of this glucose transporter primarily exerts a repressive effect on a broad spectrum of cellular functions.

KEGG pathway enrichment analysis of these DEGs provided crucial insights into the metabolic rewiring caused by the *gltp1* knockout and its direct implications for *Monascus* pigment (MP) biosynthesis (Table 2, Appendix A).

In the early fermentation stage of 36 h (S01 vs. S02 group), DEGs were significantly enriched in pathways central to primary metabolism. These included “Microbial metabolism in diverse environments,” “Pyruvate metabolism,” “Glyoxylate and dicarboxylate metabolism,” and “Glycerolipid metabolism.” The enrichment of these pathways indicates a major rerouting of central carbon flux. Pyruvate is a key hub for generating acetyl-CoA, the essential building block for the polyketide backbone of all MPs [8,14]. The glyoxylate cycle can replenish C4 metabolites, supporting energy production and biosynthesis when glucose is scarce, a condition potentially mimicked by the disrupted glucose sensing in the mutant [22].

Most notably, concerted enrichment was observed in multiple amino acid metabolism pathways, as follows: “beta-Alanine metabolism,” “Phenylalanine metabolism,” “Arginine and proline metabolism,” “Tryptophan metabolism,” “Tyrosine metabolism,” and “Valine, leucine and isoleucine degradation.” This is highly significant for MP synthesis. The degradation of branched-chain amino acids (Val, Leu, and Ile) directly supplies acetyl-CoA [8]. More importantly, amino acids like phenylalanine, tyrosine, tryptophan, and particularly arginine are known to serve as preferential amino group donors for the conversion of orange pigments to red pigments [6,9,10,24]. The transcriptional rewiring of these pathways in the *gltp1* mutant suggests a strategic cellular response to enhance the provision of both carbon skeletons and amino groups, thereby priming the metabolic network for efficient MP production, especially the red components.

By the late stage of 144 h (S03 vs. S04 group), the enrichment profile shifted dramatically towards pathways essential for cellular biosynthesis and maintenance, as follows: “Ribosome biogenesis in eukaryotes,” “Purine metabolism,” “Pyrimidine metabolism,” and “Sulfur metabolism.” This indicates a reallocation of resources to bolster the cell’s protein synthesis capacity (ribosomes) and nucleotide pools. This shift is critical for supporting the high-level expression and translation of the large enzymatic complexes required for secondary metabolism, such as the polyketide synthase (PKS) and fatty acid synthase (FAS) within the MP gene cluster. Sulfur metabolism is also crucial for the synthesis of methionine and S-adenosylmethionine [25], which are involved in various cellular methylation reactions and could influence secondary metabolite profiles. The sustained enrichment in “Microbial metabolism in diverse environments” underscores the continued global metabolic adjustment in the mutant.

Further analysis of specific genes revealed a targeted impact on signaling pathways. The expression of *gene_390199* (annotated as the G protein-coupled receptor *gprD*) was significantly downregulated in the mutant at 36 h. In *Aspergillus bombycis*, the GprD protein is part of the G-protein-coupled receptor (GPCR) complex [22]. In *Aspergillus* species, this complex has been identified to consist of nine transmembrane proteins (such as GprA-I), along with the Gα, Gβ, and Gγ subunits forming the G protein trimer, and effectors such as GTPase enzymes [22].

Concurrently, *gene_473493* (encoding Pka-C3, a catalytic subunit of the cAMP-dependent protein kinase) was significantly downregulated at 144 h (FPKM Table 3). In *Aspergillus oryzae* RIB40, Pka-C3 is a catalytic subunit of the cAMP-dependent protein kinase, which functions within the cAMP/PKA signaling pathway [22].

The literature indicates that GPCRs such as GprD (Gpr4 and Gpr1) in fungi (including *Saccharomyces cerevisiae*, *Neurospora crassa*, and *Aspergillus* species such as *Aspergillus nidulans*, as well as *Cryptococcus neoformans*) primarily function as sensors for extracellular carbon sources like glucose [22,26,27,28,29,30,31,32,33], while also serving as receptors for amino acid sensing [22,29,31]. These GPCRs further activate the cAMP/PKA signaling pathway to regulate gene expression, thereby modulating fungal growth, development, and metabolism [20,21,22,23,26,27,28,29,30,31,32,33].

Recent research by Chen et al. [21] demonstrated that knockout of the Gα protein-encoding gene in GPCRs significantly promotes *Monascus* pigment production in *Monascus* spp. fungi. Transcriptome sequencing revealed that in the Gα protein knockout strain, the expression levels of genes related to the TCA cycle, carbon-metabolism-associated MFS transporters, and nitrogen metabolism were downregulated, indicating that the Gα protein negatively regulates the synthesis of this secondary metabolite [21]. Therefore, the downregulation of MFS transporter genes involved in carbon metabolism further facilitates enhanced *Monascus* pigment production. These findings are consistent with our research results.

Our transcriptomic data support a coherent model for how GLTP1 deletion enhances MP production. The loss of GLTP1 disrupts carbon sensing, as evidenced by the suppression of the GprD receptor and the downstream cAMP/PKA pathway. This signaling defect triggers a biphasic metabolic adaptation.

The early phase (36 h) is characterized by a comprehensive reprogramming of central carbon and amino acid metabolism. This creates an abundant pool of essential precursors: acetyl-CoA from pyruvate and branched-chain amino acid degradation for the polyketide chain, and specific amino acids for the amination step that forms red pigments. The late phase (144 h) sees a shift towards enhancing the biosynthetic capacity itself, with resources funneled into ribosome and nucleotide biosynthesis to efficiently translate the MPs biosynthetic machinery. This temporal strategy—early precursor priming followed by late-stage commitment to massive enzyme production—orchestrated by the initial perturbation in carbon signaling, provides a powerful mechanistic explanation for the accelerated starch degradation and significantly higher pigment yield observed in the *gltp1* mutant.

### 3.6. Transcriptional Profiling of the Monascus Pigment Biosynthetic Gene Cluster in the gltp1 Mutant

To elucidate the molecular mechanism underlying the enhanced pigment production in the *gltp1* knockout strain *M. ruber* GLTP24, we performed a comparative transcriptomic analysis focusing on the known *Monascus* pigment biosynthetic gene cluster between the wild-type *M. ruber* CICC41233 and the *M. ruber* GLTP24 mutant at 36 h and 144 h of fermentation.

The results revealed significant differential expression of several key genes within the cluster (Figure 5, Appendix A). The polyketide synthase gene (*gene_470061*), which catalyzes the core backbone synthesis of *Monascus* pigments [1,34], showed notably lower expression in the *M. ruber* GLTP24 mutant at 36 h (FPKM: 153.34 in WT vs. 94.20 in mutant). Conversely, the gene encoding the pathway-specific transcriptional activator PigR (*gene_431934*) [35] exhibited a substantial increase in expression in the mutant at 144 h (FPKM: 73.70 in WT vs. 65.74 in mutant).

Key genes involved in the modification and maturation of pigments also displayed altered expression. The oxidoreductase gene *pigC* (*gene_497168*) was significantly downregulated in the mutant at 36 h (FPKM: 613.83 in WT vs. 298.88 in mutant). In contrast, another crucial oxidoreductase gene, *MpigH* (*gene_380063*), was markedly upregulated in the mutant at 144 h (FPKM: 1194.31 in WT vs. 3876.34 in mutant). Furthermore, the fatty acid synthase (FAS) genes (*gene_175502*, alpha subunit; *gene_416263*, beta subunit), which supply essential fatty acyl precursors for pigment synthesis [8], were downregulated in the mutant, particularly at 36 h. Interestingly, a gene encoding a Major Facilitator Superfamily (MFS) transporter (*gene_431958*) was significantly upregulated in the mutant at 144 h (FPKM: 150.16 in WT vs. 371.85 in mutant).

The transcriptional changes observed in the *M. ruber* GLTP24 mutant provide crucial insights into the regulatory role of the glucose transporter GLTP1 in *Monascus* pigment biosynthesis. The initial downregulation of the core polyketide synthase (*gene_470061*) and fatty acid synthase genes (*gene_175502*, *gene_416263*) at 36 h in the mutant may reflect a transient metabolic imbalance or a specific regulatory response to the altered carbon flux resulting from accelerated starch degradation following *gltp1* deletion [12,13].

However, the most significant changes occurred at the later fermentation stage (144 h). The upregulation of the pathway-specific transcriptional activator *pigR* (*gene_431934*) is particularly noteworthy, as its overexpression has been previously shown to dramatically enhance pigment production [35]. This suggests that the mutation in *gltp1* ultimately leads to the activation of this key regulatory switch. The strong induction of the *MpigH* oxidoreductase gene (*gene_380063*), which is involved in critical reduction steps during pigment synthesis [1,6], aligns perfectly with the experimentally observed increase in total pigment yield and the specific enhancement of yellow and red pigments in the *M. ruber* GLTP24 strain. This genetic evidence corroborates our biochemical findings that the *gltp1* knockout strain significantly increased the yields of yellow and red alcohol-soluble pigments.

The concurrent upregulation of a putative MFS transporter (*gene_431958*) in the mutant suggests a potential enhancement in the transport or secretion of pigments or their intermediates, which could be a secondary factor contributing to the higher measured pigment yield.

Collectively, these transcriptional dynamics demonstrate that the deletion of *gltp1* triggers a comprehensive reprogramming of the pigment biosynthetic gene cluster. The pattern—initial suppression of precursor-supplying pathways followed by strong late-stage activation of key regulatory (*pigR*) and biosynthetic (*MpigH*) genes—reveals a sophisticated metabolic adaptation. This adaptation likely links the perturbed carbon sensing due to the lack of GLTP1 protein, potentially through the proposed GprD-cAMP/PKA signaling pathway [21,22,23], to the enhanced synthesis of *Monascus* pigments. Thus, this study provides a direct molecular link between carbon transport, cluster-specific gene regulation, and secondary metabolite output in *M. ruber*.

## 4. Conclusions

In this study, we elucidated the pivotal role of the glucose transporter GLTP1 in regulating central carbon metabolism and secondary metabolite synthesis in *Monascus ruber*. GLTP1 is a negative regulator of pigment biosynthesis. Genetic evidence from knockout, complementation, and overexpression strains unequivocally demonstrates that GLTP1 negatively impacts *Monascus* pigment production. Its deletion is the direct cause of the hyper-production phenotype.

Enhanced pigment yield is linked to accelerated carbon utilization. The *gltp1* knockout strain *M. ruber* GLTP24 exhibited a superior capacity to degrade starch, leading to a significantly increased pool of carbon precursors that fuel the polyketide pathway for MP synthesis, an underlying signaling and metabolic reprogramming mechanism. The transcriptomic data support a model where the loss of GLTP1 perturbs the carbon-sensing mechanism, likely via the GprD-cAMP/PKA signaling pathway. This, in turn, induces a global transcriptional reprogramming that shunts carbon and nitrogen flux towards secondary metabolism, as evidenced by the significant enrichment of amino acid metabolic pathways that provide critical building blocks for pigments.

In summary, this work moves beyond the conventional view of GLTP1 as a mere nutrient transporter and establishes it as a key signaling node that couples carbon availability with the regulation of secondary metabolism. Disrupting this regulator effectively unlocks the inherent potential of *M. ruber* for high-level pigment production. These findings not only deepen our understanding of the physiological and molecular mechanisms underlying MP biosynthesis, but also provide a promising strategic basis for the metabolic engineering of industrial *Monascus* strains.

## Figures and Tables

**Figure 2 jof-11-00867-f002:**
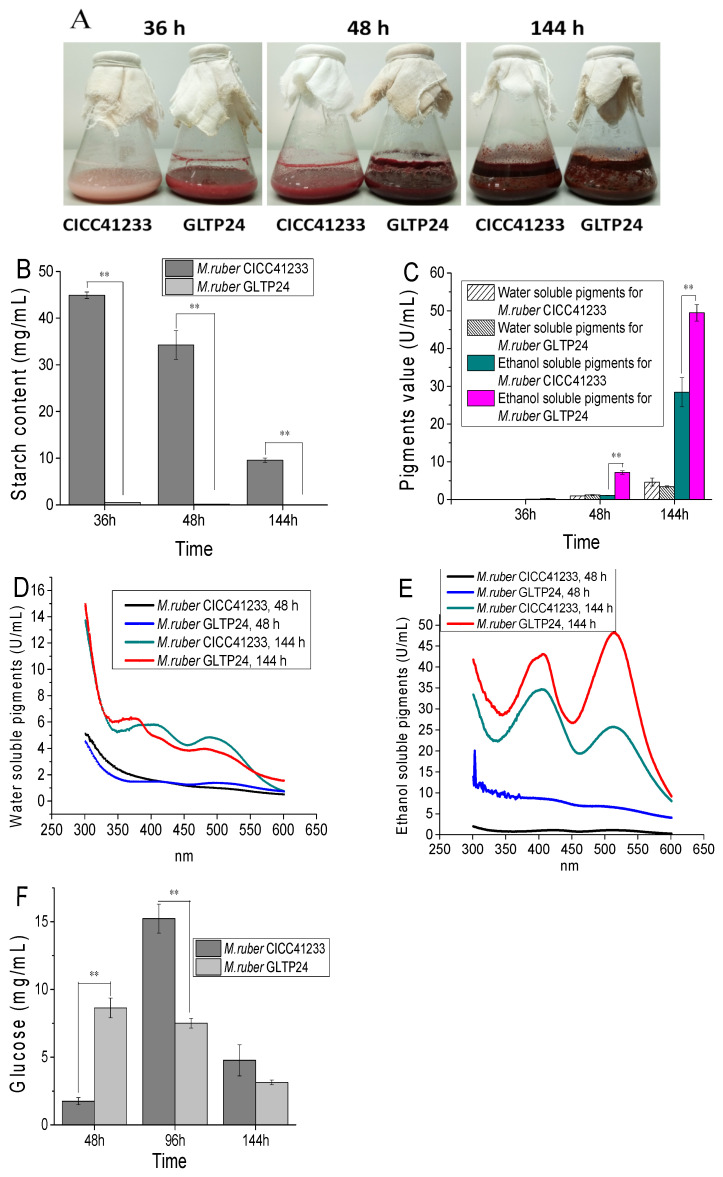
Comparative analysis of *Monascus* pigment production using *M. ruber* CICC41233 and mutant *M. ruber* GLTP24. (**A**) Phenotype of *M. ruber* at different fermentation time points; (**B**) residual starch content at different fermentation times; (**C**) color value of *Monascus* pigments; (**D**) spectral scanning of water-soluble pigments; (**E**) spectral scanning of alcohol-soluble pigments; and (**F**) glucose content in fermentation samples. ** *p* < 0.001.

**Figure 3 jof-11-00867-f003:**
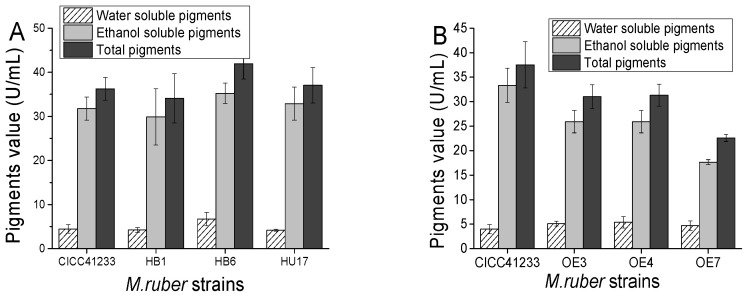
Production of *Monascus* pigments by *M. ruber* CICC41233, gltp1 gene knockout-complemented strain (**A**), and overexpressing strain (**B**) through fermentation. (**A**) The *gltp1* gene knockout-complemented strains *M. ruber* HU1, *M. ruber* HU 6, and *M. ruber* HU 17. (**B**) The *gltp1* gene overexpressing strains *M. ruber* OE3, *M. ruber* OE4, and *M. ruber* OE7.

**Figure 4 jof-11-00867-f004:**
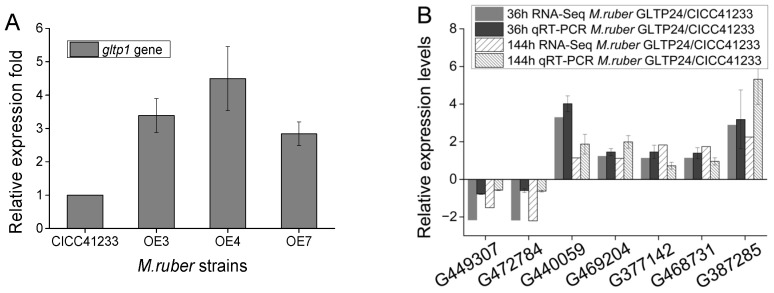
Gene expression analysis of *M. ruber*. (**A**) Relative expression fold of the *gltp1* gene in *M. ruber* OE3, OE4, and OE7, in comparison with the wild-type strain *M. ruber* CICC41233. (**B**) Relative expression levels of genes G449307, G472784, G440889, G440059, G469204, G468731, and G387285 in *M. ruber*. Error bars represent the mean ± standard deviation (SD) from four independent replicates (n = 4).

**Figure 5 jof-11-00867-f005:**
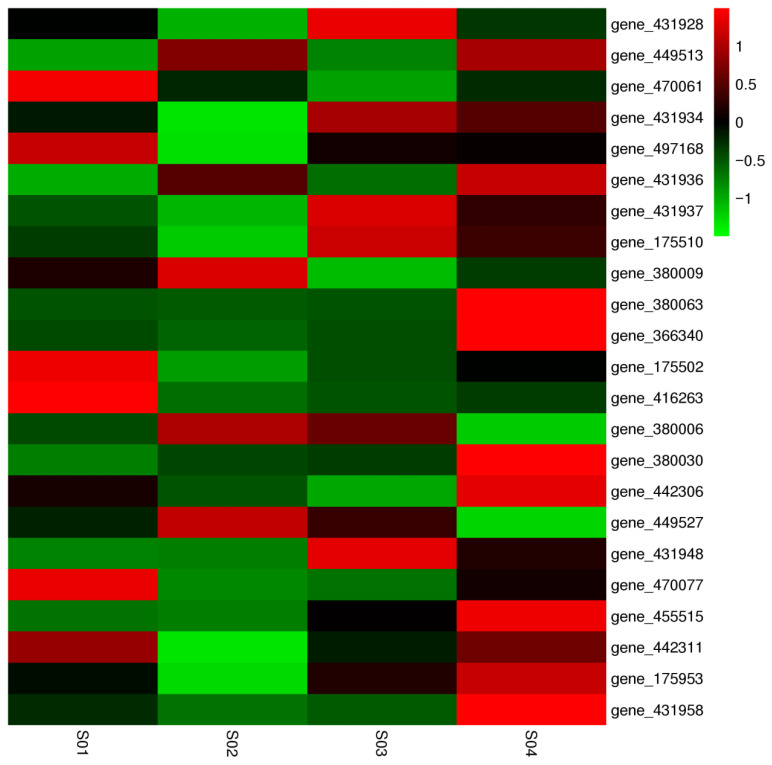
Cluster analysis of differentially expressed genes of *Monascus* pigment biosynthetic gene cluster.

**Table 1 jof-11-00867-t001:** Gene counts of differentially expressed genes (DEGs).

DEG Set *	DEG Number	Upregulated	Downregulated
S01 vs. S02	1249	515	734
S03 vs. S04	1041	374	667
S01 vs. S03	1816	811	1005
S02 vs. S04	944	346	598

* S01: 36 h sample for *M. ruber* CICC41233; S02: 36 h sample for *M. ruber* GLP24; S03: 144 h sample for *M. ruber* CICC41233; S04: 144 h sample for *M. ruber* GLP24.

**Table 2 jof-11-00867-t002:** KEGG pathway enrichment analysis of DEGs.

DEG Set	Pathway ID	KEGG Pathway	Number of Genes	*p*-Value	*q* Value
S01 vs. S02	ko01120	Microbial metabolism in diverse environments	54	0.000000	0.000007
	ko01100	Metabolic pathways	130	0.000002	0.000103
	ko00410	beta-Alanine metabolism	12	0.000008	0.000265
	ko00360	Phenylalanine metabolism	15	0.000015	0.000328
	ko00630	Glyoxylate and dicarboxylate metabolism	13	0.000017	0.000328
	ko00561	Glycerolipid metabolism	11	0.000169	0.002698
	ko00330	Arginine and proline metabolism	15	0.000401	0.005495
	ko00380	Tryptophan metabolism	14	0.000458	0.005495
	ko00350	Tyrosine metabolism	12	0.000561	0.005984
	ko00910	Nitrogen metabolism	7	0.001243	0.011935
	ko01110	Biosynthesis of secondary metabolites	56	0.001400	0.012222
	ko00053	Ascorbate and aldarate metabolism	4	0.001729	0.013835
	ko00620	Pyruvate metabolism	11	0.003274	0.024178
	ko00280	Valine, leucine and isoleucine degradation	10	0.005815	0.039873
	ko00040	Pentose and glucuronate interconversions	7	0.006760	0.043261
S03 vs. S04	ko03008	Ribosome biogenesis in eukaryotes	24	0.000000	0.000000
	ko00230	Purine metabolism	23	0.000003	0.000107
	ko00240	Pyrimidine metabolism	16	0.000167	0.004733
	ko01100	Metabolic pathways	99	0.000454	0.009638
	ko00920	Sulfur metabolism	7	0.000677	0.011514
	ko01120	Microbial metabolism in diverse environments	34	0.003339	0.047303
S01 vs. S03	ko01120	Microbial metabolism in diverse environments	59	0.000011	0.001081
	ko00051	Fructose and mannose metabolism	15	0.000253	0.009147
	ko01100	Metabolic pathways	154	0.000269	0.009147
S02 vs. S04	ko01110	Biosynthesis of secondary metabolites	47	0.000017	0.001066
	ko01100	Metabolic pathways	91	0.000023	0.001066
	ko01120	Microbial metabolism in diverse environments	33	0.000406	0.007109
	ko00670	One carbon pool by folate	6	0.000441	0.007109
	ko00630	Glyoxylate and dicarboxylate metabolism	9	0.000455	0.007109
	ko01130	Biosynthesis of antibiotics	35	0.000469	0.007109
	ko01200	Carbon metabolism	18	0.002545	0.033079

**Table 3 jof-11-00867-t003:** Gene expression profiles (FPKM) of *M. ruber* CICC41233 and *M. ruber* GLTP24.

Gene ID	Functional Annotation	36 h		144 h	
		CICC41233	GLTP24	CICC41233	GLTP24
gene_390199	G protein-coupled receptor GprD	100.31	46.97 *	30	17.74
gene_473493	serine/threonine-protein kinase (Pka-C3)	10.02	6.27	13.38	3.73 *
gene_395958	NAD(P)-binding protein	7.66	33.86 *	52.09	148.11 *

* FDR < 0.05.

## Data Availability

The raw transcriptome data are available in the NCBI database under the accession number PRJNA1370451. Data are contained within the article and Appendix A.

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
