# Peer review of "Mechanistic Insights from Transcriptomics: How the Glucose Transporter gltp1 Gene Knockout Enhances Monascus Pigment Biosynthesis in M. ruber CICC41233"

_jof, 2025, doi:10.3390/jof11120867_

Round 1
Reviewer 1 Report
English language need to be revised in the entire manuscript. It is very hard to read. The authors should ask someone fluent in English (native speaker) to check the meaning of words and sentence order. At least, there is a lot of AI tools, which can correct and fix language quite pretty
In the manuscript ‘Mechanistic Insights from Transcriptomics: How the Glucose Transporter gltp1 Gene Knockout Enhances Monascus Pigment Biosynthesis in M. ruber CICC41233‘, the authors performed the set of experiments, proving that in M. ruber gltp1 gene encoding MFS transporter responsible for glucose transport and is involved in regulating transcriptional activity of secondary metabolite-related cluster genes. The authors constructed Mrgltp1 mutated strains both, in D gltp1 background and OE gltp1-encoding MFS transporter for glucose, and they proved correlation in SM production. From the methodical point of view I have no complaints.
In my opinion, presented work is interesting and it is important to the other fungal researchers, working on Monascus species and their SMs. The carefully thought-out experiments gave the interesting results, although I see few points concerning this work, which makes it unclear, as listed below:
Row 92: citation needed of the transcriptomic study
Row 157: the gene nomenclature should be clear and available to find them among reference genomic databases, such as GeneBank and/or Uniprot. These numbers are unknown.
Row 182: this sentence miss the end. GLTP24 description is lost.
Row 205: figure numeration is chaotic. Need to be numbered and cited in proper order
Row 223: pigment secretion should be estimated in SI units (mg). Unknown units are not informative. The authors should calibrate absorbance with known standard SM and then related to secreted mg of SM.
Row 328: ...GPCR complex... citation needed
Reviewer 2 Report
The manuscript presents results on the role of the glucose transporter GLTP1 in pigment biosynthesis in Monascus ruber. The authors conducted research using mutant strains and analysing transcriptome as well as expression of several specific genes. They obtained interesting data and drew relevant conclusions. Howerever, some corrections and editing are needed to improve the expression of the study.
Lines 89-96: Please, provide reference of your "preliminary" study
Line 106: provide the origin of M. ruber CICC41233
Lines 111, 112: clarify "The gltp1 gene (https://mycocosm.jgi.doe.gov/Monru1/Monru1.home.html, Protein ID 378211, gene_129444) fragment" and provide additional details of plasmid pNeo0380
Lines 148-150: please, clarify what was used for RNA isolation ("After fermentation, 25 mL of culture broth (M.ruber CICC41233 and Δgltp1 mutant strain) was used for pigment analysis, while the rest of the sample was centrifuged (4 C, 30 min, 9,000 ×g").
Line 154: provide brief description of " Quantitative Real-Time PCR " , since the reference [14] (Line 156) is inappropriate. It refers to another paper "Long, C., M. Liu, and X. Chen. The acyl-CoA binding protein affects Monascus pigment production in Monascus ruber CICC41233. 3 Biotech 8 (2): 121.2018" where the protocol could differ (for example, "GAPDH was used as an endogenous control gene")
Line 158: provide information about primer selection (program used) or reference
Lines 166-167: clarify, what type of NCBI BLAST analysis was performed.
Please, mention in Materials and Methods Section: blast analysis and ATMT (abbreviation)
Figure 4: mark statistical significance where it is appropriate
Figure 4B: increasing size would be appreciated
Figure 5 : please, clarify. The caption mentions A and B which are absent in Figure.
Round 2
Reviewer 1 Report
The manuscript is fine. I have nothing to complain
The manuscript is fine. I have nothing to complain. The authors have referred to all issues.
Author Response
Thanks!
Reviewer 2 Report
The authors addressed some corrections. Althouth the results of qRT-PCR analysis should be re-considered and require statistical analysis.
Lines 92-99: please, mention in parentheses that these are unpublished data to clarify the source of this data.
Section "2.6. Quantitative Real-Time PCR": provide the name of the program used for primer selection or cite the appropriate references.
Figure 4: statistical significance analysis of gene expression differences must be performed, as it is mandatory for this type of analysis. Additionally , how many technical and biological replicates of the PCR reactions were performed?
Figure 4 : Please arrange the charts vertically (one below the other) to increase the image size. Despite the authors' response stating, "The necessary amendments have been made," the size remains the same.
Round 3
Reviewer 2 Report
The authors implemented most of the requested revisions. The study is valuable; however, the qPCR analysis, including the description of methodological approaches and results presentation, still requires careful editing.
1. This comment has been raised and repeated for the third time.
- Section 2.6. Quantitative Real-Time PCR: provide the name of the program used for primer selection or cite the appropriate references.
The comment is about primer design methodology, not about the list of primers. Provide the name of the software or online tool used for primer selection (such as Primer3, NCBI Primer-BLAST etc.). If a specialized program was not used or if the primers were adopted from prior studies, the appropriate references should be cited.
This detail is considered basic methodological information in PCR and, probably, the problem is misunderstanding the comment.
2. Kindly provide method used for satistical analysis in qPCR.
3. Provide additional description in Figure 4 about Error bars.
